# Effects of Health Shocks, Insurance, and Education on Income: Fresh Analysis Using CHNS Panel Data

**DOI:** 10.3390/ijerph19148298

**Published:** 2022-07-07

**Authors:** Issam Khelfaoui, Yuantao Xie, Muhammad Hafeez, Danish Ahmed, Houssem Eddine Degha, Hicham Meskher

**Affiliations:** 1School of Insurance and Economics, University of International Business and Economics, Beijing 100029, China; xieyuantao@uibe.edu.cn; 2Institute of Business and Management Sciences, University of Agriculture, Faisalabad 38040, Pakistan; hafeez_86@hotmail.com; 3School of Finance and Economics, Jiangsu University, Zhenjiang 212013, China; danish.ahmed88@live.com; 4School of Foreign Language, Shanghai Jianqiao University, Shanghai 201315, China; 5Department of Business Administration, HANDS—Institute of Development Studies (HANDS-IDS), Karachi 75230, Pakistan; 6Center for Islamic Finance, University of Bolton, Bolton BL3 5AB, UK; 7International Institute on Governance and Strategy (IIGS), Beijing 100000, China; 8Department of Mathematics and Computer Science, Faculty of Science and Technology, Université de Ghardaia, Ghardaia 47000, Algeria; degha.houssem@outlook.com; 9Laboratory of Valorization and Promotion of Saharan Resources (VPSR), Kasdi-Merbah University, Ouargla 30000, Algeria; hicham.meskher@g.enp.edu.dz

**Keywords:** health shocks, income, insurance, education

## Abstract

The most important asset for a person is their health and wellbeing. While it is possible to keep one’s health at its best, it is common for people to have health shocks (HSs; negative shocks to an individual’s health). In this study, using Chinese Health and Nutrition Survey (CHNS) panel data, we studied the impact of different HSs on income using new modified methods. Thus, we considered the substantial links among different HSs, levels of education, and insurance types, as well as their impact on people’s wealth defined by their income. The core aim of this study is to help devise and guide new policies to reduce the effect of these HSs through the proper use of education and insurance channels. In this research, we used simple pooled OLS regression to measure the different causality estimates of HSs, education, and insurance, as well as their interactions. Obtained through the use of up-to-date panel data, the study results are consistent with previous research using different HS and education measures. The findings of this research suggest revising previous policies concerning education levels and health insurance schemes.

## 1. Introduction

In daily life, health and wealth represent the most important assets for an individual, whereby having both at their best assures a smooth and healthy life. Although both are important, losing health is comparatively worse than losing wealth for several reasons, as a person can more easily recover from losing wealth. Losing health affects not only a person’s wellbeing but also their welfare. In the modern economy, case studies of health loss to varying degrees, such as the unpredictable occurrences of health shocks (HSs), have been a common research direction.

HSs represent one of the most significant and common risks facing individuals in a society. Several previous researchers identified HSs in several ways, whether by using a proxy (injuries, accidents, acute illnesses, death, or their combination) or an objective measure such as the number of days a person was not able to carry out their activities [1]. HSs can also be subjectively evaluated through a change in self-assessment proxies whether compared to peers or to one’s previous status. There are also other types of HSs, such as those related to mental health and social health. In the short term, HSs damage an individual’s health and wealth due to the high cost of treatment, time cost, irregular income pattern, interruption of children’s education, and workforce absence [2]. Low-income individuals are relatively more affected by HSs [2]. When parents are experiencing HSs, children may be required to work or stay at home to create extra income or to simply take care of the ill parent when one or both parents are sick [3,4,5]. In the long term, these HSs incur additional costs for an individual [6].

Economic individuals use several strategies to cope with such occurrences, such as saving or seeking help from other family members [7]. The most common strategy used by individuals and governments is *insurance*. Health insurance is considered a safe way to hedge, to a great degree, against the adverse effects of these shocks more effectively in the short term [8]. On an individual level, taking the example of the effect of HSs on expenditures by individuals/households, insurance was found to have a mild positive effect on the outcome and minimize the overall cost [1,7].

Our aim was to contribute to the existing research by examining the relationship among different types of HSs (mediated by insurance and education) on income. This study evaluated the most up-to-date data on Chinese rural individuals and their interaction with various HS variables mediated by insurance and education. The investigation of different HSs and their interaction terms accounts for the novelty presented in this paper, with a focus on Chinese rural areas, unlike previous studies conducted in the United States and Europe.

The HSs in our research were grouped as subjective and objective shocks. This grouping was inspired by previous studies, although they mostly used a proxy of an HS to determine its causal effect on income. Examples include the number of days spent unable to carry out normal activities (objective measure) [1] and the self-perceived view of one’s health and wellbeing (subjective measure). The interactions of different HSs with insurance and education are usually evaluated separately, with substantial evidence indicating that insurance mitigates the loss of income due to HSs or health problems. It is also widely known that better-educated people tend to make healthier choices and have fewer unhealthy habits, thereby reducing the severity of HSs. Our research inspects the interaction of all three variables with different HSs.

This study was based on panel data from the Chinese Health and Nutrition Survey (CHNS) [9]. This survey contains several variables, such as education, income, healthcare, insurance, and children. Involving households from rural and provincial China, data collection has been conducted since 1989 in waves of 3 years over 15 provinces. The data currently consist of more than 200,000 individuals belonging to more than 11,000 households. The choice to use this data source was made for multiple reasons: the length of the collection period (more than 15 years); its collection from the same households and individuals over this period; numerous local reforms and new policies throughout the collection period (involving areas such as education, pension funds, agriculture, health, labor, and child labor); the occurrence of the global economic crisis; a large number of observations; and the survey’s completion by Chinese authorities.

Our results found a significant negative effect of HSs on a person’s income. This negative effect was reduced through insurance and education, whereby a good level of both factors could result in a reduction in or elimination of this effect. These results were especially significant in the short term, while a long-term framework needs to be established. A strong policy of maintaining a certain level of insurance and education is recommended for rural individuals, facilitated by the establishment of a special insurance scheme for rural individuals to minimize HS occurrences. Lastly, a health policy is also needed to reduce the extra hidden health expenses resulting from these HSs.

The remainder of this paper is organized as follows: a literature review in Section 2, a description of the empirical model in Section 3, a presentation of the empirical results in Section 4, and a conclusion in Section 5.

## 2. Literature Review

HSs and their effects on income have been a part of several previous studies, many of which identified the causal effect of HSs on human capital through the consumption of labor supply and wealth or income. The aim of our study was to inspect this causal relationship, mediated by both education and insurance.

The major importance of health and human capital in the labor market is well known [10,11]; Grossman constructed a model of the demand for “good health”. In his model, to achieve better outcomes in terms of human lifespan and to achieve great consumption of goods and leisure throughout this period, a high level of health must be maintained. Better outcomes can be achieved when the cost of maintaining this level is at its lowest, the level of human capital is at its highest (in terms of better wages and job experience), and the level of education is high. Some models for predicting health were also developed in [12,13,14]. These models had the same assumption that a starting stock of health is inherited by individuals, which increases with investments in health and decreases over time. On a microlevel, it should be considered that individuals make rational decisions regarding health and related investments. Thus, if health curves are maintained and are predictable through good health, then individuals tend to make decisions accordingly. Furthermore, no effect on labor supply and productivity occurs, indicating that an individual’s wealth is not affected by their health. However, an unpredicted occurrence affecting health, negative in nature, which alters these curves may greatly affect an individual’s wealth (i.e., reduce their income through absence or work inefficiency). In other words, the Grossman model of health demand has the goal of optimizing an individual’s inputs to achieve their best utility. Thus, to achieve better wealth outcomes, investments in health are needed. The resulting utility is described in terms of wealth as a function of health: wealth = F(health; other non-health inputs).

### 2.1. Introducing HSs into the Wealth Equation

Models of the consequences of health outcomes on labor wages often assume an inelastic labor supply and focus on the effects on wages, capacity to work, time in labor, and other socioeconomic factors [12,13,14,15,16,17,18].

Empirically, in the United States, many studies used the Retirement History Survey [14,19,20,21,22,23,24]. For example, the authors of [19] stated that income labor supply, future choice, and human capital are influenced by deviations in health. How much a person can work on a job is proportionally related to how healthy they are, which determines the level of skill and experience they can obtain related to the job. Thus, a negative deviation in health, i.e., a health shock, may disrupt the previous course of events. The authors of [19] studied the interaction between human capital and health through their models by specifically assessing the interaction of health and wages and the biased outcomes of such an interaction. For instance, many occurrences of HSs may force older workers into an early retirement [20,21]. Wealth is also decreased through medical expenses due to ill health [22,23,24]. In the United States, the effect of health and health shocks on income was determined using various types of data [17,25,26,27,28]. Some general effects of health on income were revealed using the National Health Interview Survey (NHIS) [27]. Wealth is also decreased through medical expenses due to ill health [25,26]. The authors of [17] also found a more severe negative effect of HSs on workers who did not attend college, with respect to their saving strategy and wealth accumulation.

In Britain, many occurrences of HSs may force older workers into an early retirement [29]. According to a paper analyzing the role of health and employment, using data from the first 12 waves of the British Household Panel Survey (1991–2002), many occurrences of HSs pushed younger individuals to quit the labor market, indicating how work limitations can affect income [30]. Another paper using the British Household Panel Survey provided evidence for self-reported health shocks and early retirement by the British [31]. Similarly, in Canada, older workers are forced into an early retirement due to many incidences of HSs [32]. Additionally, for British households, HSs result in a permanent reduction in consumption levels [33]. In Spain and Germany, HSs push younger individuals to quit the labor market, which reduces their income through limited work presence [34,35]. Furthermore, using Dutch hospital and tax register data, the authors of [36] found that the effects of HSs on employment are persistent, whereby the causal effects of sudden illness, represented by acute hospitalizations, on employment and income remained for up to 6 years after the health shock. On the basis of a random 10% sample of Danish citizens from the period 1981–2000, the authors of [37] identified the short- and long-term effects of health shocks (road accidents and injuries) on income, wages, and employment for male individuals.

According to evidence from Indonesia and Ethiopia, sudden sickness, ill health, and HSs all create long-term economic consequences for individuals. The most prevalent direct economic cost is medical care, whereas the most prevalent indirect cost is the loss in wages due to the effect of reduced labor supply and productivity [38,39,40]. In Vietnam, the unpredictable occurrence of HSs can lead to poverty traps and loss of wages [41]. The effect of HSs is transferred from parents to children, greatly affecting their education. In Ethiopia, to cope with a parent’s ill health (especially in the case of long-term and severe HSs) and to supplement the lost income or to replace the productivity of the parent, early child labor, work at home (extra family chores), and an increased incidence of missing school are widely noticed [38]. For South Korean households, taking care of sick individuals also places another economic burden on healthy individuals, which is related to residential mobility [42]. In India and South Africa, the authors of [43] found a strong bidirectional causal link between income and health.

In China, using panel data from Chinese farm households covering the period of 1987–2002, the authors of [44] studied how major HSs for household adults affect children’s schooling. Utilizing a subjective measure of HSs, calculated using data from four rounds of the CHNS from 1989 to 2000, the authors of [45] were the first to study their effect on income, consumption, and labor market outcomes. These are the closest to our research and are the only studies investigating the effect of HSs on the income of Chinese individuals before 2000. In this regard, the first gap in the literature appears, whereby most previous studies only considered the effect of one type of HS on different wealth variables. Furthermore, very few studies were conducted in China, with the only studies evaluating panel data up to the year 2000 (exactly four waves) [45]. Many changes have occurred in the Chinese health, education, and insurance systems since the year 2000, and at least six more waves have been added to the panel. An important geographical and time-related gap, therefore, arises in the study of the effects of different HSs on the income of Chinese individuals. In this section, we also observed that studies in the Chinese setting were more household-focused than individually focused, and they did not consider various types of HSs. To reduce the ambiguity of the relationship between HSs and income, the wealth utility function was transformed to have wealth levels as the output of a sudden change in health status and other socioeconomic variables (e.g., the prevalence of HSs), where income = F(HS; other non-health inputs).

### 2.2. Introducing Insurance into the Equation

For the sake of public policymaking, it is important and necessary to understand the different mechanisms taken by individuals to cope with HSs [16,46,47]. To cope with HSs, one major proposition is insurance [16,46]. According to [48], there was no obvious significant impact of HSs on the income of uninsured groups compared to insured groups. Using general health status, i.e., health self-assessment, as a proxy for HSs, the authors of [45] used similar Chinese data to the current research to not only study the impact of such shocks on the income of households, their labor supply, and their health expenditures, but also to examine the effect of insurance on economic growth. They deduced the same results for the negative effect of HSs on wages, household income, and labor supply; surprisingly, the negative effect was stronger for insured households compared to uninsured households. Furthermore, urban households may suffer more from HSs compared to rural ones [41]. Having health and disability insurance is a better situation, and not having it, due to uncertainties in income, may greatly affect households [49]. It was also found that the expected retirement age is affected by perceived health status, according to a probit model developed by [31]. The authors of [50], using data from European Union countries, proposed that the establishment of social security insurance reduced the severity of the negative effects of HSs on labor supply. According panel data from Bengal, households will make different agricultural choices and acquire small debts to cope with losses due to HSs [51]. In Mexico, foreign remittances were used to offset the effects of HSs [52]. Using a smaller sample size of the same Chinese panel data as this study, the authors of [1] provided empirical evidence on the role of public health insurance in offsetting the negative outcomes associated with HSs by minimizing their effects on household income and consumption. In this regard, considering findings from the literature, the wealth utility function was transformed to have wealth levels determined by HSs, mediated by insurance, where income = F(HS(insurance); other non-health inputs). Accordingly, a third gap in the literature appears, as previous studies did not consider the mediating effect of insurance on the health of Chinese individuals (specifically with regard to HSs) and the resulting outcomes for their income and wealth.

### 2.3. Introducing Education into the Equation

It is also known that education and health are two pivotal determinants of human capital, and they have a complex relationship and a strong correlation even when controlling for other demographic and socioeconomic variables [53,54]. Multiple policies have been suggested with the aim of raising the levels of both education and health [53,55,56,57]. It is clear that such policies will only be effective when considering both determinants, thereby affecting personal income, wealth, and labor supply [53,56,57]. Using different health measures has led to different causal results, with some validating the causal link between health and education [54,58,59,60,61,62,63,64,65], while others failed to find such evidence [53,56,57,66,67,68]. According to the literature, to the best of our knowledge, only two studies investigated the effect of education on health in China [58,59]. Although our aim was not to study the causal link between education and health, it is important to evaluate their relationship to determine the interaction of HSs, insurance, and education. In this regard, considering findings from the literature, the wealth utility function was transformed to have wealth levels determined by HSs, mediated by education and insurance, where income = F(HS(insurance and education); other non-health inputs). Accordingly, a fourth gap in the literature appears, as previous studies did not consider the mediating effects of both insurance and education on the health shocks of Chinese individuals and the resulting outcomes for their income and wealth.

From the literature review, we can clearly see that previous studies typically investigated one type of HS, using objective or subjective measures, and determined its effect on one economic variable. Our research takes into consideration both objective and subjective HS variables to determine any changes in their causal effects. Furthermore, the mediating effects of education and insurance on the results were evaluated.

## 3. Materials and Methods

### 3.1. Sample Set

Our data were sourced from the Chinese Health and Nutrition Survey (CHNS) [9]; the survey results were obtained through a random, multistage cluster process, tracked over time, with a sample size of 215,352 individuals. The CHNS has information on multiple individual and household-level demographics, such as insurance channel, health, income, and education. So far, eight waves of the survey have been conducted since 1989, with the latest update coming in 2015. The survey data were collected every 2–3 years; we used the entire dataset in this study, focusing on health, education, and insurance. The survey was conducted across 9–11 provinces differing in terms of economic development. We calibrated the data to include individuals eligible to work, i.e., aged 18 to 60 years, following a previous study [1]. Our data were fairly obtained through Chinese government sites, such as the official open-access site of the CHNS [9].

### 3.2. Variable Definition

#### 3.2.1. Dependent Variable: Individual Wealth

To measure individual wealth, we used the log monthly income from the CHNS data as our dependent variable.

#### 3.2.2. Independent Variables

##### Health Shocks

Our HS variable was divided into objective HSs and subjective HSs, as mentioned in [11], with slight changes, as outlined in Table 1.

#### 3.2.3. Control Variables

We included county number, marital status, sex, work experience, household number, and other demographic variables to control our estimations.

### 3.3. Econometric Strategy and Model Specification

On the basis of the model first used by [45], our approach applied pooled OLS regression to determine the effects of *HS*, insurance, and education on income.
(3)$$INC_{it}=\alpha_{0}+\alpha_{1}HS_{it}+\alpha_{2}\;Edu_{it}+\alpha_{3}\;INS_{it}+\beta_{1}^{\prime }IS_{it}+\beta_{2}^{\prime }X_{it}+\varepsilon_{it},$$
(4)$$\beta_{1}^{’}IS_{it}=\gamma_{1}HS_{it}\times INS_{it}+\gamma_{2}HS_{it}\times Edu_{it}+\gamma_{3}HS_{it}\times INS_{it}\times Edu_{it},$$
(5)$$HS_{ijt}=\pi \;SHS_{it}+1-\pi \;OHS_{it},$$
where $INC_{it}$ is the log average monthly income of individual *i* in year *t;* $HS_{it},\;Edu_{it},\;\mathrm{and}\;INS_{it}$ are health shocks (subjective and objective HSs), education level, and insurance, respectively; $IS_{it}$ is the set of interaction terms explained in Equation (4); $\varepsilon_{it}$ is the error term; and $X_{it}$ is the set of control variables.

The estimation faced three challenges. Firstly, potential measurement errors in the HS variable may appear [69]. Self-perceived health status may be influenced by the health system, education, and other factors, resulting in measurement errors in subjective measures of HSs, in contrast to objective measures of HS [1] such as change in BMI or illness status [21,70,71], the appearance of a new serious health condition between waves of panel data [16,23,48], the onset of impairment or disability [72,73,74], or a change in daily living activities or the number of days unable to work [11,39]. Using the same approach as [45], we focused on changes in subjective HSs across different waves, thereby mitigating measurement errors. In this way, time-invariant measurement errors in subjective HSs were eliminated. Secondly, in our paper, we did not consider the presence of unobserved individual heterogeneity, which may be a determinant of both health and income. This would have biased the OLS estimates. We overcame this problem by following the same approach as previous papers, using the panel dimensions of the data to estimate a fixed-effect model. Lastly, estimation problems can arise if health is simultaneously determined using income and health inputs. In our framework, flows or changes in health over time reflected investment in health, depreciation of health stock, and unexpected shocks. A correlation between the error term and HS variable was then obtained. The simultaneity problem is more severe when using subjective HSs compared to objective HSs. To our knowledge, no instrumental variable exists to adequately resolve this issue. While it is an imperfect method, we considered the degree of shock to address this issue.

## 4. Results

### 4.1. Investigation of Variables

Focusing on the dependent outcome variable, i.e., monthly average wages over the last year, from Table 2, we can see that the average was RMB 200 per individual, indicating that some individuals had almost no income over the last year; furthermore, this value was surprisingly under the Chinese national average, which may have been due to the selection of individuals from rural areas and provinces.

Our most important independent variable, HSs, was deduced from three aspects. Firstly, the average hospitalization time was no more than 1 day for Chinese individuals in rural and provincial areas. The results also indicated that at the provincial level, Chinese individuals did not miss many days of work, which might be due to the overall culture of great health awareness compared to other cultures. Alternatively, one HS variable may not provide a good enough picture of the health status of individuals. We found a relatively small average value for subjective HSs (SHSs) as a result of the way in which they were calculated.

The education level of Chinese individuals was seemingly low, as most were local farmers without any formal education. On average, individuals had 2 years of education compared to a maximum of 9 years.

Our results revealed that only one-third of the population had medical insurance, mostly cooperative insurance, despite the government providing insurance for all individuals. This indicates that most individuals either do not use it or do not know of its existence. Government-paid insurance accounted for only 3–5% of the sample, while private medical insurance accounted for only 2%.

The control variables exhibited a similar effect to previous studies, and they did not affect the research outcome, as they were controls; thus, they served only as a checkpoint and are not discussed.

### 4.2. Empirical Results of HS Variables

#### 4.2.1. Objective HSs (OHSs)

OHSs were measured according to the number of days an individual was unable to continue daily activities and/or number of days hospitalized in the last 4 weeks. On the other hand, the ratio of hospitalized days lost 4 weeks over the same period of 28 days. The model was investigated as a function of the level of education, the use of insurance, and their joint effect on OHSs with respect to an individual’s income. The results are presented in Table 3 and Table 4, where each column indicates a variation of the model.

In the first model, the occurrence of health shocks (OHS_U or OHS_H) negatively affected income, in line with previous studies, with an increase in OHS_H by one unit resulting in up to a 14% loss of income, while an increase in OHS_U by one unit resulted in a 25–45% loss of income. Following a simple health shock, the days that an individual spends unable to carry out daily activities may be 2–4 times greater than the number of days spent at the hospital. Furthermore, changes in OHS_U may be related to the method of determining the level of education, whereby attaining a high level of education (25%) had a fivefold greater impact than the number of years of formal schooling (5%).

A novelty of this research is that we added the interaction term of OHS_U to a high education level to determine the mediating effect of good education on HSs affecting income. The results showed that a good education (8%) could lead to a threefold reduction in the effect of health shocks (24%) on income. Furthermore, having medical insurance has a very small but negative impact on income, with Chinese individuals paying a small annual cost to obtain medical insurance [1]. However, wealthier individuals may opt for special medical insurance, with the negative effect on income being more obvious according to the number of insurances held by an individual.

In the second model, we introduced the variable of whether the illness led to admission to hospital. While this did not affect OHS_H, it greatly reduced the OHS_U variable from 25% to approximately 6–8%. This indicates that a great proportion of individuals choose not to visit the hospital when sick due to dislike or a lack of trust. Alternatively, this could indicate that individuals visiting the hospital are cured more quickly, resulting in a faster reattainment of productivity and income.

In the third model, we focused on the effect of insurance to determine the source of the negative effect of having multiple types of insurance. This may be due to other variables in the model contributing to the effect, as they are indirectly included in the definition of having medical insurance. We found that having commercial insurance had a positive impact, while having government-provided insurance, whether medical insurance or the New Rural Cooperative Medical Scheme (NCMS), had a negative impact. We propose that the reason for this difference is rural and county-level individuals not knowing how to use insurance, despite paying a subsidy. In contrast, wealthier individuals typically opt for commercial insurance and know how to use it, getting good coverage for their sick days. Another reason, as mentioned by [1], is that the old medical scheme failed miserably to cover rural and county-level individuals, which led to the new scheme being put in place. Individuals could still be paying the dividend for the old scheme, while individuals using the new scheme may not know how to use it or may not trust it.

The fourth model sheds light on the increase in OHS_U, whereby having definite medical insurance had no effect or a small negative effect on income, with the other parameters remaining significant. In the fifth model, we took a deeper look at the interaction of education, OHSs, and insurance. Firstly, the negative effect of having medical insurance was reduced for educated people, especially for those who completed years of education only in regular school as compared to those who attained high-level education. More educated people may tend to have multiple insurances as an extra precaution, resulting in an additional premium. Furthermore, the results indicated that more educated people were not as affected by health shocks, which may be related to better life choices. Despite the negative estimator obtained for the interaction of education and OHSs with insurance, having insurance and being more educated seemed to reduce the effect of HSs on income.

#### 4.2.2. Subjective HSs (SHSs)

As in the previous table, Table 5 measures the impact of SHSs on income. Compared with OHSs, the occurrence of SHSs decreased income more severely on average. This may be explained by the fact that how individuals feel about their health has more of an effect on their absence from the workforce. Hence, a more negative change in self-perceived status results in a greater loss of income, which is worsened by a low level of education. However, education reduced the negative significance of having medical insurance, thus positively reducing the effect of health shocks on income.

### 4.3. Weighted Average HSs

In this section, we estimated the value of HSs through a weighted average of OHSs and SHSs by varying the weight parameter π to measure the effect on income. The most significant parameters can then be considered for policymaking and creating insurance models while considering education level (πSHS + 1 − πOHS). As a starting point, we considered one HS variable as being more important in reflecting the occurrence of illness and/or injury and its effect on income. However, we can see from the graphs for the estimation (Figure 1) of HSs that neither SHSs nor OHSs can be neglected, with the value of π = 0.5 indicating their equal importance, despite the fivefold greater negative impact of SHSs on income. This may be due to their causal relationship, whereby one or multiple variables can be omitted if we determine their linearity. With respect to interaction terms, the conclusion remains that an insured, educated individual would not be affected by the occurrence of HSs, whether considering SHSs, OSHs, or both.

## 5. Conclusions

In this study, the main goal was to investigate how health shocks affect an individual’s income by analyzing Chinese Health and Nutrition Survey (CHNS) panel data from rural areas. Several aspects were considered, including education, income, healthcare, insurance, and children. The main focus was on insurance and education, with the effects of these two aspects being inversely proportional. The authors determined the effect of health shocks, mediated by insurance and education, on income for individuals in urban and rural Chinese areas. It can be concluded that education and insurance have an inversely proportional mediating effect on the impact of HSs on income. Moreover, subjective and objective HSs affect the income of Chinese rural and urban individuals differently.

Our findings suggest the importance of updating policies from a government perspective concerning insurance and education to elevate health protection. Different proxies can be used to measure health in an attempt to elevate the health status of individuals in rural and urban areas. Our study adds to the extensive research direction relating an individual’s health to their wealth, along with establishing possible causality. Future research should consider group heterogeneity, long-term causality, and possible time effects. Future studies can also consider using other measures for health and wealth.

## Figures and Tables

**Figure 1 ijerph-19-08298-f001:**
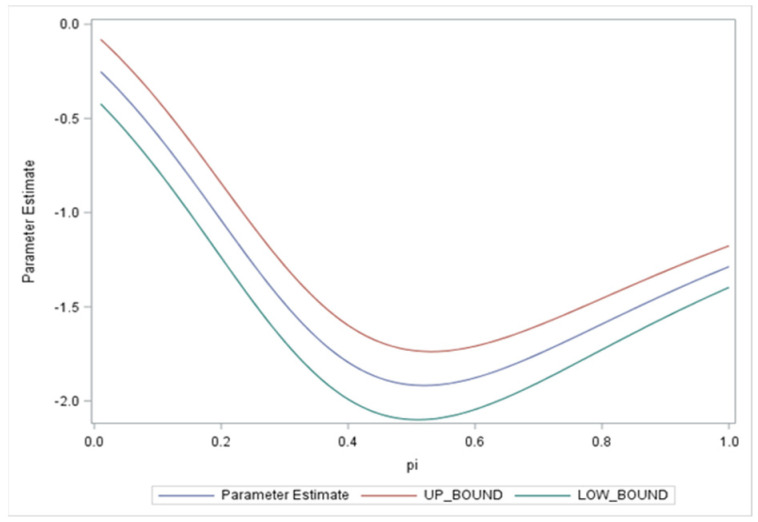
Upper and lower bands of HS parameter estimation for simulated values of π between 0 and 1.

**Table 1 ijerph-19-08298-t001:** Measurement methods of health shocks.

Subjective HS	Objective HS
(1) $$SHS_{t,i}=\frac{\min\left({\mathrm{SR}}_{t,i}-{\mathrm{SR}}_{t-1,i},0\right)}{-4}$$	(2) $$OHS_{t,i}=\frac{SD_{t,i}}{28}$$
${\mathrm{SR}}_{t,i}$: self-reported assessment (very good = 5 to bad = 0), according to the CHNS data; t: year of recording; t − 1: previous year or previous wave according to the data; i: individual.	$SD_{t,i}$: sick days, days of hospitalization, or days where a person was unable to carry out their normal activities in the last 4 weeks; t: wave year; i: individual.

Education: Education was measured using the following proxies: completed years of formal education in regular school and highest level of education attained. Insurance: From the CHNS dataset, several dummy variables were used to evaluate insurance, such as use of medical insurance, insurance type, and commercial vs. governmental health insurance.

**Table 2 ijerph-19-08298-t002:** Presents a description of the variables.

Variable Coding	Label	*N*	Mean	SD	Min	Max
INC	Average monthly wage last year	215,352	200.111	3420.88	0	999,999
OHS_U	Unable to carry out normal activities in the last 4 weeks	215,352	0.0100	0.0930	0	3.5357
IOC	Inpatient or outpatient care	215,352	0.0072	0.0845	0	1
OHS_H	Days hospitalized in the last 4 weeks	215,352	0.0037	0.1049	0	28.6785
SHS	Self-reported assessment	215,352	−0.2047	0.6589	−4	0
CEDU	Completed years of formal education in regular school	151,139	17.548	8.9965	0	36
HEDU	Highest level of education attained	154,014	1.5516	1.3778	0	9
INS	Use of medical insurance	215,352	0.3333	0.4714	0	1
INST	Insurance type	215,352	0.2884	0.4530	0	1
INSC	Insurance type: commercial	215,352	0.0211	0.1439	0	1
INSUFM	Insurance type: urban free medical	215,352	0.0555	0.2289	0	1
INSRCB	Insurance type: rural cooperative basic	215,352	0.1537	0.3607	0	1
ADW	Average number of days per week worked last year	215,352	1.3924	2.5117	0	9
PW	Presently working	215,352	0.4079	0.4915	0	2
SLN	Spouse’s line number	215,352	2.8625	14.606	0	181
MLH	Mother living in household	215,352	0.3053	0.4605	0	1
FLH	Father living in household	215,352	0.2858	0.4518	0	1
PROV	Province	215,352	38.790	9.6323	11	55
URSR	1 = urban site (U); 2 = rural site (R)	215,349	1.6906	0.4622	1	2
UNCN	U: 1–2 = city number; 1–4 = county number	215,349	2.2026	1.0801	1	4
UR	U: 1–2, 5–6, 9–10 = urban; 3–4, 7–8, 11–12 = suburban; 1, 5, 9 = town; 2–4, 6–8, 10–12 = village	215,349	2.7594	1.3681	1	9
HN	Household number	215,349	21.4240	30.6238	1	180
MSI	Money spent on illness or injury	215,352	14.1004	650.5129	0	88,888
AGE	Calculated age in years to 0 decimal points	213,405	33.6580	21.5426	0	101
G	Gender	215,298	0.4942	0.4999	0	1

**Table 3 ijerph-19-08298-t003:** Parameter estimations for first set of objective health shocks.

Parameter	(1)	(2)	(3)	(4)	(5)	(6)	(7)	(8)	(9)	(10)	(11)	(12)
**OHS_U**	**−0.434 *** (−3.59)**	**−0.256 *** (−3.53)**			**−0.218 *** (−3.48)**	**−0.207 *** (−3.19)**					**−0.145 ** (−2.39)**	**−0.114 * (−1.86)**
**OHS_ H**			**−0.143 ** (−2.37)**	**−0.142 ** (−2.39)**			**−0.062 (−1.14)**	**−0.065 (−1.22)**	**−0.439 *** (−3.61)**	**−0.275 * (−1.77)**		
**HEDU**	**0.245 *** (54.42)**			**0.246 *** (54.82)**	**0.236 *** (55.84)**			**0.236 *** (55.89)**	**0.287 *** (40.90)**			**0.289 *** (41.29)**
**CEDU**		**0.052 *** (51.71)**	**0.052 *** (51.81)**			**0.049 *** (53.06)**	**0.049 *** (53.16)**			**0.056 *** (39.86)**	**0.057 *** (40.23)**	
**INS**	**−0.063 *** (−3.28)**	**−0.071 *** (−3.53)**	**−0.072 *** (−3.59)**	**−0.064 *** (−3.34)**	**−0.003 (−0.20)**	**−0.003 (−0.17)**	**−0.004 (−0.23)**	**−0.004 (−0.26)**	**0.000 (1.51)**	**0.000 (1.55)**	**0.000 (1.51)**	**0.000 (1.52)**
**INST**	**−0.886 *** (−33.37)**	**−0.871 *** (−30.25)**	**−0.868 *** (−30.15)**	**−0.884 *** (−33.27)**					**−0.741 *** (−22.82)**	**−0.671 *** (−13.57)**	**−0.664 *** (−13.43)**	**−0.737 *** (−22.71)**
**INSC**					**1.546 *** (3.83)**	**1.479 *** (3.61)**	**1.477 *** (3.61)**	**1.545 *** (3.83)**				
**INSUFM**					**−1.032 *** (−41.55)**	**−1.020 *** (−37.80)**	**−1.018 *** (−37.73)**	**−1.030 *** (−41.47)**				
**INSRCB**					**−0.518 *** (−7.42)**	**−0.474 *** (−6.22)**	**−0.472 *** (−6.20)**	**−0.517 *** (−7.40)**				
**OHS_U_ HEDU**	**0.082 * (1.73)**								**0.153 ** (2.57)**			**Not significant**
**INS_HEDU**									**−0.067 *** (−7.75)**			**−0.068 *** (−7.87)**
**INS_HEDU _ OHS_U**									**−0.120 ** (−2.14)**			**−0.083 ** (−2.24)**
**INS_CEDU**										**−0.009 *** (−4.94)**	**−0.009 *** (−5.08)**	
**CEDU_OHSH**										**0.010 (1.22)**	**Not significant**	
**INS_CEDU_ OHS_H**										**−0.016 ** (−2.41)**	**Not significant**	
MSI	0.000 (1.43)	0.000 (1.50)	0.000 (1.47)	0.000 (1.41)	0.000 ** (2.15)	0.000 ** (2.28)	0.000 ** (2.25)	0.000 ** (2.12)	−0.087 *** (−4.46)	−0.086 *** (−4.22)	−0.086 *** (−4.23)	−0.087 *** (−4.50)
FLH	0.109 ***	0.136 *** (5.65)	0.137 *** (5.68)	0.110 *** (4.74)	0.069 *** (3.34)	0.093 *** (4.30)	0.093 *** (4.32)	0.070 *** (3.36)	0.111 *** (4.78)	0.136 *** (5.66)	0.137 *** (5.69)	0.111 *** (4.80)
MLH	0.134 *** (6.02)	0.132 *** (5.71)	0.131 *** (5.69)	0.134 *** (6.00)	0.134 *** (6.72)	0.123 *** (5.96)	0.123 *** (5.94)	0.133 *** (6.70)	0.129 *** (5.82)	0.130 *** (5.62)	0.129 *** (5.57)	0.129 *** (5.80)
SLN	0.009 *** (33.56)	0.009 *** (32.72)	0.009 *** (32.72)	0.009 *** (33.55)	0.006 *** (27.32)	0.007 *** (26.94)	0.007 *** (26.95)	0.007 *** (27.33)	0.009 *** (33.47)	0.009 *** (32.69)	0.009 *** (32.72)	0.009 *** (33.46)
PW	0.574 *** (11.79)	0.507 *** (9.57)	0.507 *** (9.58)	0.575 *** (11.81)	0.455 *** (10.44)	0.400 *** (8.43)	0.400 *** (8.43)	0.455 *** (10.45)	0.556 *** (11.41)	0.493 *** (9.29)	0.492 *** (9.29)	0.557 *** (11.43)
ADW	−0.111 *** (−26.26)	−0.121 *** (−27.98)	−0.121 *** (−27.94)	−0.111 *** (−26.21)	−0.119 *** (−31.40)	−0.126 *** (−32.32)	−0.126 *** (−32.29)	−0.119 *** (−31.36)	−0.110 *** (−26.09)	−0.121 *** (−27.96)	−0.121 *** (−27.88)	−0.110 *** (−26.04)
G	0.042 *** (3.56)	−0.002 (−0.17)	−0.002 (−0.19)	0.041 *** (3.54)	0.094 *** (8.97)	0.057 *** (5.24)	0.057 *** (5.24)	0.094 *** (8.97)	0.042 *** (3.57)	−0.002 (−0.20)	−0.003 (−0.22)	0.041 *** (3.54)
AGE	0.034 *** (53.92)	0.036 *** (53.19)	0.036 *** (53.16)	0.034 *** (53.87)	0.028 *** (48.04)	0.029 *** (47.77)	0.029 *** (47.73)	0.028 *** (47.99)	0.034 *** (53.97)	0.036 *** (53.23)	0.036 *** (53.21)	0.034 *** (53.95)
PROV	−0.011 *** (−19.53)	−0.012 *** (−19.76)	−0.012 *** (−19.68)	−0.011 *** (−19.46)	−0.007 *** (−14.02)	−0.007 *** (−13.79)	−0.007 *** (−13.73)	−0.007 *** (−13.96)	−0.011 *** (−19.77)	−0.012 *** (−19.95)	−0.012 *** (−19.86)	−0.011 *** (−19.73)
USRS	0.185 *** (14.41)	0.170 *** (12.76)	0.171 *** (12.78)	0.185 *** (14.40)	0.076 *** (6.48)	0.072 *** (5.90)	0.073 *** (5.92)	0.076 *** (6.49)	0.182 *** (14.16)	0.166 *** (12.45)	0.166 *** (12.45)	0.182 *** (14.14)
UNCN	0.042 *** (7.16)	0.044 *** (7.15)	0.044 *** (7.17)	0.043 *** (7.17)	0.020 *** (3.81)	0.022 *** (3.96)	0.022 *** (3.98)	0.020 *** (3.84)	0.042 *** (7.18)	0.044 *** (7.16)	0.044 *** (7.19)	0.042 *** (7.18)
UR	0.150 *** (41.41)	0.152 *** (40.05)	0.152 *** (40.03)	0.150 *** (41.38)	0.106 *** (32.44)	0.106 *** (30.51)	0.106 *** (30.51)	0.107 *** (32.44)	0.149 *** (41.19)	0.152 *** (39.76)	0.151 *** (39.75)	0.149 *** (41.14)
HN	0.009 *** (52.13)	0.008 *** (50.32)	0.008 *** (50.33)	0.009 *** (52.14)	0.005 *** (36.42)	0.005 *** (35.15)	0.005 *** (35.17)	0.005 *** (36.45)	0.008 *** (51.99)	0.008 *** (50.28)	0.008 *** (50.31)	0.008 *** (51.98)
Intercept	3.343 *** (47.84)	2.887 *** (36.99)	2.882 *** (36.93)	3.338 *** (47.77)	3.990 *** (63.38)	3.510 *** (49.96)	3.506 *** (49.90)	3.987 *** (63.32)	3.283 *** (46.72)	2.808 *** (35.20)	2.799 *** (35.12)	3.278 *** (46.65)
Adj_R^2^	0.4737	0.4684	0.4683	0.4735	0.5803	0.5745	0.5744	0.5801	0.4747	0.4689	0.4687	0.4746

Note: Standard deviations are indicated in parentheses below the coefficients as percentages; *, **, and *** represent significance at the 10%, 5%, and 1% levels, respectively. The parameters in bold highlight the independent variables estimates targeted by the model.

**Table 4 ijerph-19-08298-t004:** Parameter estimations of the second set of objective health shocks.

Parameter	(13)	(14)	(15)	(16)	(17)	(18)	(19)	(20)
IOC_OHS_H	−0.069 (−0.78)	−0.087 (−0.96)					−0.062 (−1.15)	−0.063 (−1.18)
IOC_OHS_U			−0.144 ** (−2.38)	−0.141 ** (−2.36)	−0.065 (−0.83)	−0.076 (−0.94)		
HEDU	0.246 *** (54.83)			0.246 *** (54.82)	0.236 *** (55.89)			0.236 *** (55.89)
CEDU		0.052 *** (51.83)	0.052 *** (51.81)			0.049 *** (53.16)	0.049 *** (53.16)	
INS	−0.065 *** (−3.37)	−0.073 *** (−3.61)	−0.072 *** (−3.59)	−0.064 *** (−3.35)	−0.004 (−0.27)	−0.004 (−0.23)	−0.004 (−0.23)	−0.004 (−0.26)
INSC					1.546 *** (3.83)	1.478 *** (3.61)	1.477 *** (3.61)	1.545 *** (3.83)
INSUFM					−1.031 *** (−41.51)	−1.019 *** (−37.76)	−1.018 *** (−37.73)	−1.030 *** (−41.47)
INSRCB					−0.517 *** (−7.41)	−0.472 *** (−6.20)	−0.472 *** (−6.20)	−0.517 *** (−7.40)
INST	−0.885 *** (−33.32)	−0.869 *** (−30.20)	−0.868 *** (−30.15)	−0.884 *** (−33.27)				
FLH	0.110 *** (4.75)	0.137 *** (5.69)	0.137 *** (5.68)	0.110 *** (4.74)	0.070 *** (3.36)	0.093 *** (4.32)	0.093 *** (4.32)	0.070 *** (3.36)
MSI	0.000 (1.41)	0.000 (1.47)	0.000 (1.47)	0.000 (1.41)	0.000 ** (2.12)	0.000 ** (2.25)	0.000 ** (2.25)	0.000 ** (2.12)
MLH	0.133 *** (5.99)	0.131 *** (5.68)	0.131 *** (5.69)	0.134 *** (6.00)	0.133 *** (6.70)	0.123 *** (5.94)	0.123 *** (5.94)	0.133 *** (6.70)
SLN	0.009 *** (33.56)	0.009 *** (32.73)	0.009 *** (32.72)	0.009 *** (33.56)	0.007 *** (27.33)	0.007 *** (26.95)	0.007 *** (26.95)	0.007 *** (27.33)
PW	0.575 *** (11.80)	0.507 *** (9.56)	0.507 *** (9.58)	0.575 *** (11.81)	0.454 *** (10.44)	0.399 *** (8.42)	0.400 *** (8.43)	0.455 *** (10.45)
ADW	−0.111 *** (−26.21)	−0.121 *** (−27.94)	−0.121 *** (−27.94)	−0.111 *** (−26.21)	−0.119 *** (−31.36)	−0.126 *** (−32.28)	−0.126 *** (−32.29)	−0.119 *** (−31.36)
G	0.042 *** (3.58)	−0.002 (−0.16)	−0.002 (−0.19)	0.041 *** (3.54)	0.094 *** (8.99)	0.057 *** (5.26)	0.057 *** (5.24)	0.094 *** (8.97)
AGE	0.034 *** (53.85)	0.036 *** (53.15)	0.036 *** (53.16)	0.034 *** (53.87)	0.028 *** (47.99)	0.029 *** (47.73)	0.029 *** (47.73)	0.028 *** (47.99)
PROV	−0.011 *** (−19.48)	−0.012 *** (−19.70)	−0.012 *** (−19.68)	−0.011 *** (−19.46)	−0.007 *** (−13.98)	−0.007 *** (−13.74)	−0.007 *** (−13.73)	−0.007 *** (−13.97)
USRS	0.185 *** (14.42)	0.171 *** (12.80)	0.171 *** (12.78)	0.185 *** (14.40)	0.076 *** (6.50)	0.073 *** (5.93)	0.073 *** (5.92)	0.076 *** (6.49)
UNCN	0.043 *** (7.18)	0.044 *** (7.17)	0.044 *** (7.17)	0.043 *** (7.18)	0.020 *** (3.84)	0.022 *** (3.98)	0.022 *** (3.98)	0.020 *** (3.84)
UR	0.150 *** (41.43)	0.153 *** (40.07)	0.152 *** (40.03)	0.150 *** (41.38)	0.107 *** (32.46)	0.106 *** (30.52)	0.106 *** (30.51)	0.107 *** (32.44)
HN	0.009 *** (52.15)	0.008 *** (50.34)	0.008 *** (50.33)	0.009 *** (52.14)	0.005 *** (36.45)	0.005 *** (35.16)	0.005 *** (35.17)	0.005 *** (36.45)
Intercept	3.338 *** (47.76)	2.881 *** (36.92)	2.882 *** (36.93)	3.338 *** (47.77)	3.987 *** (63.32)	3.506 *** (49.90)	3.506 *** (49.90)	3.987 *** (63.32)
Adj_R^2^	0.4735	0.4682	0.4683	0.4735	0.5801	0.5744	0.5744	0.5801

Note: Standard deviations are indicated in parentheses below the coefficients as percentages; **, and *** represent significance at the 5%, and 1% levels, respectively.

**Table 5 ijerph-19-08298-t005:** Parameter estimations in the models for subjective health shocks and simulated weighted HSs.

Parameters	(21)	(22)	(23)	(24)	*p* = 0.1	*p* = 0.01	*p* = 0.5
**SHS**	**−0.220 *** (−23.45)**	**−0.195 *** (−20.49)**	**−0.322 *** (−22.87)**	**−0.345 *** (−21.27)**			
**WHS**					**−1.286 *** (−22.87)**	**−0.2518 *** (−2.88)**	**−1.9150 *** (−20.41)**
CEDU		0.051 *** (51.63)		0.055 *** (39.18)			
HEDU	0.246 *** (55.29)		0.282 *** (40.64)		**0.2816 *** (40.64)**	**0.2880 *** (41.21)**	**0.2814 *** (40.53)**
INS	−0.051 *** (−2.68)	−0.059 *** (−2.95)	−0.064 *** (−3.32)	−0.057 *** (−2.85)	**−0.0639 *** (−3.32)**	**−0.0860 *** (−4.43)**	**−0.0622 *** (−3.22)**
INST	−0.861 *** (−32.65)	−0.845 *** (−29.51)	−0.746 *** (−23.17)	−0.706 *** (−14.38)	**−0.7455 *** (−23.17)**	**−0.7395 *** (−22.79)**	**−0.7475 *** (−23.21)**
**INS _HEDU**			**−0.048 *** (−5.53)**		**−0.0479 *** (−5.53)**	**−0.0679 *** (−7.85)**	**−0.0536 *** (−6.21)**
**INS _HEDU_SHS**			**0.055 *** (9.96)**				
**INS _HEDU_WHS**					**0.2206 *** (9.96)**	**−0.0333 *** (−0.74)**	**0.2678 *** (6.90)**
**INS _CEDU**				**−0.005 *** (−2.96)**			
**INS _CEDU_SHS**				**0.009 *** (11.53)**			
MSI	0.000 (1.51)	0.000 (1.55)	0.000 (1.53)	0.000 (1.57)	0.000 (1.53)	0.000 (1.53)	0.000 (1.61)
FLH	0.105 *** (4.58)	0.133 *** (5.57)	0.107 *** (4.65)	0.132 *** (5.51)	0.1066 *** (4.65)	0.1108 *** (4.78)	0.1053 *** (4.58)
MLH	0.129 *** (5.86)	0.127 *** (5.55)	0.119 *** (5.42)	0.119 *** (5.18)	0.1194 *** (5.42)	0.1291 (5.81)	0.1239 *** (5.62)
SLN	0.009 *** (33.37)	0.009 *** (32.60)	0.009 *** (33.36)	0.009 *** (32.69)	0.0087 *** (33.36)	0.0088 *** (33.47)	0.0087 *** (33.36)
PW	0.565 *** (11.70)	0.503 *** (9.55)	0.550 *** (11.40)	0.494 *** (9.40)	0.5503 *** (11.4)	0.5561 *** (11.42)	0.5507 *** (11.39)
ADW	−0.112 *** (−26.65)	−0.122 *** (−28.38)	−0.111 *** (−26.59)	−0.122 *** (−28.46)	−0.1113 *** (−26.59)	−0.1100 *** (−26.06)	−0.111 *** (−26.53)
G	0.039 *** (3.34)	−0.004 (−0.34)	0.040 *** (3.43)	−0.003 (−0.23)	0.0396 *** (3.43)	0.0413 *** (3.55)	0.0389 *** (3.36)
AGE	0.033 *** (52.10)	0.035 *** (51.59)	0.033 *** (52.07)	0.035 *** (51.54)	0.0329 *** (52.07)	0.0343 *** (53.97)	0.0333 *** (52.73)
PROV	−0.011 *** (−20.17)	−0.012 *** (−20.32)	−0.011 *** (−20.06)	−0.012 *** (−20.22)	−0.0111 *** (−20.06)	−0.011 *** (−19.77)	−0.011 *** (−20.18)
USRS	0.176 *** (13.82)	0.162 *** (12.22)	0.174 *** (13.68)	0.160 *** (12.09)	0.1742 *** (13.68)	0.1818 *** (14.15)	0.1749 *** (13.71)
HN	0.009 *** (52.73)	0.008 *** (50.89)	0.042 *** (7.12)	0.043 *** (7.12)	0.0084 *** (52.38)	0.0084 *** (51.98)	0.0084 *** (52.26)
UR	0.145 *** (40.44)	0.148 *** (39.07)	0.144 *** (40.18)	0.147 *** (38.74)	0.1442 *** (40.18)	0.1490 *** (41.17)	0.1449 *** (40.31)
UNCN	0.042 *** (7.14)	0.044 *** (7.20)	0.008 *** (52.38)	0.008 *** (50.53)	0.0417 *** (7.12)	0.0423 *** (7.16)	0.0414 *** (7.05)
Intercept	3.407 *** (49.11)	2.952 *** (38.04)	3.340 *** (47.92)	2.869 *** (36.26)	3.3397 *** (47.92)	3.2805 *** (46.69)	3.3424 *** (47.88)
Adj_R^2^	0.4820	0.4752	0.4842	0.4777	0.4747	0.4827	0.4842

Note: Standard deviations are indicated in parentheses below the coefficients as percentages; and *** represents significance at the and 1% levels, respectively. The parameters in bold highlight the independent variables estimates targeted by the model.

## Data Availability

The datasets used and/or analyzed during the current study are available from the corresponding author on reasonable request.

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
