# Peer review of "Effects of Health Shocks, Insurance, and Education on Income: Fresh Analysis Using CHNS Panel Data"

_ijerph, 2022, doi:10.3390/ijerph19148298_

Round 1

Reviewer 1 Report

  1. References 1, 66-73, 83, 89 are not sighted in the text
  2. p.5  ....kai (2016) ..... is this last name? 
  3. p.5  ....kai (2016) ..... p.9. ..... Liu (2016) ..... why you use in this way references. They differ from the way you used the other references.
  4. Conclusions too short. You need to revise it.

Author Response

Reply to Reviewers Comments for Paper ijerph-1673745

Effects of Health Shocks, Insurance, and Education on Income: Fresh Analysis Using CHNS Panel Data

Dear Editor and Reviewers,

We would like to commence by thanking the editor and the three reviewers for their valuable time and constructive comments. Their expert knowledge of the field has helped us to strengthen the manuscript significantly. According to the valuable suggestions provided by the reviewers, we have revised the manuscript. We endeavored to address all the comments and our reflections are now given below point by point.

Sincerely,

The Authors

Reviewer1

Comments and Suggestions for Authors

  1. References 1, 66-73, 83, and 89 are not sighted in the text
  2. p.5  ....kai (2016) ..... is this last name? 
  3. p.5  ....kai (2016) ..... p.9. ..... Liu (2016) ..... why you use in this way references. They differ from the way you used the other references.

Response: The authors appreciate the comments provided by the honored reviewer. In this version, the authors have carefully enhanced the paper's concerns of all the referees. To remedy the issue mentioned in comments 1, 2, and 3, we applied the referencing style of the journal through Zotero; and have made the necessary corrections. And to keep the consistency of the referencing style, we have removed the mention of both Liu (first name) and Kai (author last name) and kept only their reference number.

  1. Conclusions too short. You need to revise it.

Response: The authors appreciate the comments provided by the honored reviewer. In this version, the authors have carefully revised the conclusion and wrote it in the following way

“In this study, the main goal was to investigate how health shocks affect an individual’s income by analyzing Chinese Health and Nutrition Survey (CHNS) panel data from rural areas. Several aspects were considered, including education, income, healthcare, insurance, and children. The main focus was on insurance and education, with the effects of these two aspects being inversely proportional. The authors determined the effect of health shocks, mediated by insurance and education, on income for individuals in urban and rural Chinese areas. It can be concluded that education and insurance have an inversely proportional mediating effect for HS on income. Moreover, subjective and objective HSs affect the income of Chinese rural and urban individuals differently.

Our findings suggest the importance of updating policies from a government perspective concerning insurance and education to elevate health protection. Different proxies can be used to measure health in an attempt to elevate the health status of individuals in rural and urban areas. Our study adds to the extensive research direction relating an individual’s health with their wealth, along with possible causality. Future research should consider group heterogeneity, long-term causality, and possible time effects. Future studies can also consider using other measures for health and wealth.”

Reviewer 2 Report

The main goal of this paper is to study how Health Shocks affect people’s income by analyzing the Chinese Health and Nutrition Survey (CHNS) panel data from Chinese rural areas. The authors have considered several aspects, including education, income, health care, insurance, children, and so on. Among these variables, the authors mainly focus on insurance and education, and they think the effects of health shocks and these two aspects are inversely proportional.

The topic is interesting, and the chosen panel data set is authoritative and representative. However, as stated in the introduction, the authors have emphasized their goal is to study how HS affects people’s income, but this manuscript is basically introducing the relationship between health shocks, income, insurance, and education. 

Author Response

Reply to Reviewers Comments for Paper ijerph-1673745

Effects of Health Shocks, Insurance, and Education on Income: Fresh Analysis Using CHNS Panel Data

Dear Editor and Reviewers,

We would like to commence by thanking the editor and the three reviewers for their valuable time and constructive comments. Their expert knowledge of the field has helped us significantly strengthen the manuscript. According to the valuable suggestions provided by the reviewers, we have revised the manuscript. We endeavored to address all the comments and our reflections are now given below point by point.

Sincerely,

The Authors

Reviewer2

Comments and Suggestions for Authors

The main goal of this paper is to study how Health Shocks affect people’s income by analyzing the Chinese Health and Nutrition Survey (CHNS) panel data from Chinese rural areas. The authors have considered several aspects, including education, income, health care, insurance, children, and so on. Among these variables, the authors mainly focus on insurance and education, and they think the effects of health shocks and these two aspects are inversely proportional.

The topic is interesting, and the chosen panel data set is authoritative and representative. However, as stated in the introduction, the authors have emphasized their goal is to study how HS affects people’s income, but this manuscript is basically introducing the relationship between health shocks, income, insurance, and education. 

Response: The authors appreciate the comments provided by the honored reviewer. In this version the authors have carefully enhanced the paper's concerns of all the referees To maintain consistency, we changed the statement on the gaol in the introduction and the conclusion which is basically introducing the relationship between health shocks, income, insurance, and education. As we see that the relationship in this direction is still new and many new findings could be made especially since the results of both education and insurance reforms all around the world and in China are only giving fruits in these years, especially for the same level of income individuals (mostly low-income individuals). As per the manuscript in general we have sent it for MDPI English editing services to correct the language as a whole and make it much more readable and appealing.

Reviewer 3 Report

Basically, the authors are strong at methods.  However, the literature review part is not well-organized. The readers could not see the hypothsis of the relationship between health shock, education, insurance, and income. As the authors wrote, the relationship between them are complex and the readers are at a loss and don't know why you put up with the hypothesis. In addition, the noverlty of this paper is not that strong and quite similar studies have been published. 

Author Response

Reply to Reviewers Comments for Paper ijerph-1673745

Effects of Health Shocks, Insurance, and Education on Income: Fresh Analysis Using CHNS Panel Data

Dear Editor and Reviewers,

We would like to commence by thanking the editor and the three reviewers for their valuable time and constructive comments. Their expert knowledge of the field has helped us to strengthen the manuscript significantly. According to the valuable suggestions provided by the reviewers, we have revised the manuscript. We endeavored to address all the comments and our reflections are now given below point by point.

Sincerely,

The Authors

Reviewer 3

Comments and Suggestions for Authors

Basically, the authors are strong at methods.  However, the literature review part is not well-organized. The readers could not see the hypothesis of the relationship between health shock, education, insurance, and income. As the authors wrote, the relationship between them are complex and the readers are at a loss and don't know why you put up with the hypothesis. In addition, the novelty of this paper is not that strong and quite similar studies have been published. 

Response: The authors appreciate the comments provided by the honored reviewer. In this version the authors have carefully enhanced the papers concerns of all the referees. In this new version, we have reorganized the literature review part. Where we divide it into three parts. We started by giving a brief literature on the conceptual utility function where the consumers want to maximise their wealth outcome through maintaining better health inputs. Then, in the first part, we illustrated the reconstruction of the function with specifying from the literature the sources that measured wealth outcome is income and health as health shocks. In this same part we argued that there is both a geographical and time gap of an econometric evaluation (maximisation) of this utility of less health shocks to better income or more health shocks to less income causal relationship. Then in the second part we introduced, through the literature, insurance as a variable or a mediator of health, and showed that studies didn’t consider it inside the dynamics of health shocks on income. And finally, in the third part, we added education to the equation also as a variable of health. In this manner we clarified for the readers the hypothesis of the relationship between health shock, education, insurance, and income. As for the novelty, the way we constructed, the variables, and the econometric approach of this paper is only considered in our design plus there is huge gap on showing the effects of different HS on the income of Chinese individuals especially after 2000, where similar studies have been published show mostly before 2000 or study it differently focusing only on one aspect. Many changes after 200 and reforms were don on the Chinese education and insurance schemes, therefore the strength is apparent in using all the available panel data waves. Especially including the last six, 2000 to 2015, would depict much better the causal relationship of HS and income with the interactions of Education and Insurance.

Round 2

Reviewer 3 Report

The authors did a comprehensive revision. Now the paper looks much better.